# Evaluation of insecticide treated window curtains and water container covers for dengue vector control in a large-scale cluster-randomized trial in Venezuela

**Audrey Lenhart**[1], **Carmen Elena Castillo**[1,2], **Elci Villegas**[2], **Neal Alexander**[3], **Veerle Vanlerberghe**[4], **Patrick van der Stuyft**[4,5], **Philip J. McCall**[1]*

**1** Vector Biology Department, Liverpool School of Tropical Medicine, Liverpool, United Kingdom, **2** Universidad de los Andes, Núcleo Rafael Rangel, Instituto Experimental Jose Witremundo Torrealba, Trujillo, Venezuela, **3** MRC International Statistics and Epidemiology Group, London School of Hygiene & Tropical Medicine, London, United Kingdom, **4** Department of Public Health, Institute of Tropical Medicine, Antwerp, Belgium, **5** Department of Public Health and Primary Care, Ghent University, Ghent, Belgium

* philip.mccall@lstmed.ac.uk

## Abstract

### Background

Following earlier trials indicating that their potential in dengue vector control was constrained by housing structure, a large-scale cluster-randomized trial of insecticide treated curtains (ITCs) and water jar covers (ITJCs) was undertaken in Venezuela.

### Methods

In Trujillo, Venezuela, 60 clusters (6223 houses total) were randomized so that 15 clusters each received either PermaNet insecticide-treated window curtains (ITCs), permanent insecticide-treated water storage jar covers (ITJCs), a combination of both ITCs and ITJCs, or no insecticide treated materials (ITMs). A further 15 clusters located at least 5km from the edge of the study site were selected to act as an external control. Entomological surveys were carried out immediately before and after intervention, and then at 6-month intervals over the following 27 months. The Breteau and House indices were used as primary outcome measures and ovitrap indices as secondary. Negative binomial regression models were used to compare cluster-level values of these indices between the trial arms.

### Results

Reductions in entomological indices followed deployment of all ITMs and throughout the trial, indices in the external control arm remained substantially higher than in the ITM study arms including the internal control. Comparing the ratios of between-arm means to summarise the entomological indices throughout the study, the combined ITC+ITJC intervention had the greatest impact on the indices, with a 63% difference in the pupae per person indices between the ITC+ITJC arm and the internal control. However, coverage had fallen

**Data Availability Statement:** All relevant data are within the manuscript and its Supporting information files.

**Funding:** This research was supported under the European Union, DENCO project, 'Towards successful dengue prevention and control' (INCO-CT-2004-517708). NA receives support from the Medical Research Council UK (Grant Reference MR/R010161/1) and PJM's research on peri-domestic behavior of Aedes aegypti receives support from MRC-UK (MR/T001267/1). The funders had no role in study design, data collection and analysis, decision to publish, or preparation of the manuscript.

**Competing interests:** The authors have declared that no competing interests exist.

below 60% by 14-months post-intervention and remained below 40% for most of the remaining study period.

## Conclusions

ITMs can impact dengue vector populations in the long term, particularly when ITCs and ITJCs are deployed in combination.

## Trial registration

ClinicalTrials.gov ISRCTN08474420; www.isrctn.com.

## Author summary

Dengue is a serious mosquito-borne disease and a threat to an estimated one third of the human population throughout the tropics. Prevention and control of dengue outbreaks is limited to vector control, and most public health programs use a variety of methods to kill the primary mosquito vector, *Aedes aegypti*. Water holding containers harboring the mosquito's immature stages can be treated or eliminated, in addition to control measures that target infected adult mosquitoes. Sustainable interventions that effectively target adult mosquitoes are needed to increase the options for control of dengue and other *Aedes*-borne viral diseases. The use of insecticide-treated curtains (ITCs) has previously been shown to significantly reduce *Ae. aegypti* numbers in and around homes, but the impact of insecticide-treated jar covers (ITJCs) is less known. The results of this study demonstrated that both ITCs and ITJCs can reduce entomological indices, with the greatest impact detected when they were deployed together.

## Introduction

With a third of the global population at risk of infection, and an annual toll of nearly 400 million infections, dengue has become one of the most intractable public health challenges of recent decades [1,2]. Despite considerable progress, neither safe effective vaccines nor prophylactic drugs suitable for mass protection are available and vector control is still the only option for the prevention or control of dengue outbreaks. However, *Aedes aegypti* is uniquely adapted to its human host and is remarkably difficult to control in the urban environments where it thrives and where an ever-increasing majority of the human population lives today [3].

Control strategies in dengue endemic countries typically include activities for reducing the abundance of potential sites where the immature stages can develop, and insecticide treatment of sites that cannot be eliminated. Adult mosquitoes are targeted by periodic space spraying, usually outdoors, or residual treatments at breeding habitats or indoor resting sites. Despite community education and often high levels of community participation, success has been limited and sustainability is a challenge. Without reliable strategies for preventing or responding to outbreaks, control programs lack efficacy and the magnitude of the global dengue public health problem has continued to grow [4–7].

Most control measures target the immature stages of the mosquito, as many of the larval habitats are comparatively easy to identify and remove or treat with insecticide. Interventions

targeting immature stages have limited impact, as at best, they can impact only vector density. Interventions targeting adult mosquitoes have an immediate effect by killing adult mosquitoes infected with dengue and can impact both vector density and longevity, and thus stand to have a greater effect on disease transmission. However, the only widely employed adult dengue vector control intervention is insecticidal space spraying, which can be appropriate when used in focal outbreak situations [8] but has only temporary and minimal impact [4], and is not a viable or sustainable long-term control method.

Insecticide-treated materials (ITMs) such as window curtains and water storage container covers have considerable potential as sustainable frontline interventions to target adult dengue vectors and have been evaluated in a range of studies in different contexts. Many of these trials [9–13] demonstrated that ITMs could be effective in reducing dengue vector densities rapidly to levels that could impact dengue transmission. Two studies [9,10] also demonstrated a spill-over effect, such that control houses located closer to houses using ITMs were less likely to have had vector infestations than those further away. In contrast, no consistent impacts on vector populations were reported in randomised-controlled trials of insecticide-treated window curtains (ITCs) and water jar covers (ITJCs) in Thailand [14], and ITCs in Peru [15] and Cuba [16,17]. In the studies from Cuba, the lack of marginal effect on vector infestation levels and on disease incidence has been attributed to the epidemiological context, characterised by long-standing and intensive routine *Aedes* control activities and already low levels of dengue transmission. The Peruvian study was compromised first, by faulty ITCs, which lost efficacy and required manual re-treatment and re-installation after only 6 months of deployment, and second, by the emergence of pyrethroid resistance before the trial ended. These trials also revealed the importance of house design as a determinant of ITM impact. The majority of the older and traditional houses in the former locations were constructed from multiple plywood panels, many of which were ill-fitting, damaged or removable, with numerous gaps allowing mosquito entry, exit and movement between houses. In Peru, many buildings had high open eaves, lacked ceilings and shared roof spaces with neighbours, while in Thailand, many modern houses were designed to open the entire ground floor to the exterior. With so many alternative access points, impacts of ITCs hung in windows and doors would be minimal. As pyrethroid-treated nets are only weakly repellent and require direct contact with mosquitoes to repel or kill, nets covering windows and doors should form a physical barrier to mosquito entry into the house. Hence, ITCs installed as fitted screens on windows and doors of intact buildings can rapidly reduce indoor mosquito densities to significantly low levels that are sustained after the insecticide treatment is lost [18,19].

Here we report on a large cluster-randomized controlled trial evaluating ITCs and ITJCs in northern Venezuela in dengue-endemic communities where the housing and appeared to be suitable for deployment.

Dengue is a major public health problem in Venezuela, the country from where the highest incidence of DHF in infants in Latin America has previously been reported [20]. At the time of this study in 2007 and 2008, Venezuela reported more than 50% of the total reported cases of both classic and hemorrhagic or severe dengue in the Andean sub-region [21,22]. In Venezuela, as in most other dengue-endemic countries, methods targeting adult mosquitoes are desperately needed to suppress vector populations below dengue transmission thresholds. Encouraging results were obtained in earlier studies evaluating ITMs in Venezuela [9], on the basis of which two large-scale cluster randomized trials were planned and completed. The first of these trials was in Thailand [14] and the second, in Venezuela, is reported here.

## Methods

### Ethics statement

This study received approval from the Institutional Review Boards (IRBs) at the Liverpool School of Tropical Medicine (Ref. 06.12; 02/02/2006) and the bio-ethical committee of the José Witremundo Torrealba Research Institute at the University of the Andes, Trujillo, Venezuela (18/06/2006).

The trial was registered with the International Standard Randomized Controlled Trial Register: ISRCTN08474420. Verbal consent was obtained for ITC deployment and entomological monitoring activities, as approved by all IRBs. Written consent was obtained for all blood draws from a parent or guardian since participants were < 18 years of age.

### Study area

The study site was located in the foothills of the Venezuelan Andes, in Trujillo State. Trujillo State has a mean annual rainfall of 750 mm, with 2 seasonal peaks in April and October. Temperatures range between 16–37˚C, with the hottest period occurring between July and September and the coldest period between December and February [23]. Earlier studies in this area reported that large domestic water drums or barrels, 150-200litre in size, were producing, together with discarded tires, buckets and tanks, an estimated 70% of adult *Ae. aegypti* in the dry season and 80% in the wet season [24]. The majority of houses were concrete structures with relatively small windows which could be covered by ready-made curtains of fixed dimensions. Hence, the site was considered suitable for deployment of ITMs designed to target eclosion, oviposition and endophily in *Ae. aegypti*. The study area included 5 suburban parishes (Monay, Flor de Patria, Pampán, Pampanito and Motatán) (Fig 1) which were selected from 3 municipalities based on their sharp increases in dengue incidence in the 3 years preceding the start of the study [24]. The study began with household recruitment and a baseline survey in July 2006 and ran through April 2009, when all follow-up surveys were completed.

### Study design

The study was undertaken as a cluster-randomised controlled trial, across 60 clusters with each cluster occupying an area greater than 500 x 500 m and comprising between 50 and 100 houses. Sample sizes were calculated according to the methodology described by Hayes and Bennett [25] and had a power of 80% to detect a 7-fold decrease in the Breteau index at an alpha error level of 0.05 (assuming a between-cluster coefficient of variation of 0.50). Allocation to treatment arm was assigned randomly: 15 clusters received insecticide-treated curtains (ITCs), 15 clusters received insecticide-treated covers for water storage jars/drums (ITJCs), 15 received both ITCs and ITJCs and a control group of 15 clusters received no intervention. Randomization of clusters to study arms was conducted by a simple lottery whereby members of the study team drew pieces of paper out of a sack that contained papers with the codes corresponding to each potential study cluster. All occupied households were eligible for inclusion while business-only premises and apartment buildings of >2 storeys were excluded. These clusters were not separated by buffer zones. An additional 15 clusters located at least 5km from the edge of the study site in the parish of Monay were selected to act as an 'external control' (Fig 1). These were selected based on similarities to the study clusters in dengue incidence from the preceding 2 years, household density and socio-demographic characteristics. These clusters were monitored in exactly the same way as the study clusters.

Recruitment and baseline surveys occurred in June 2006, interventions were placed in July-August 2006, and 5 follow-up surveys occurred in September 2006 (1 month post-

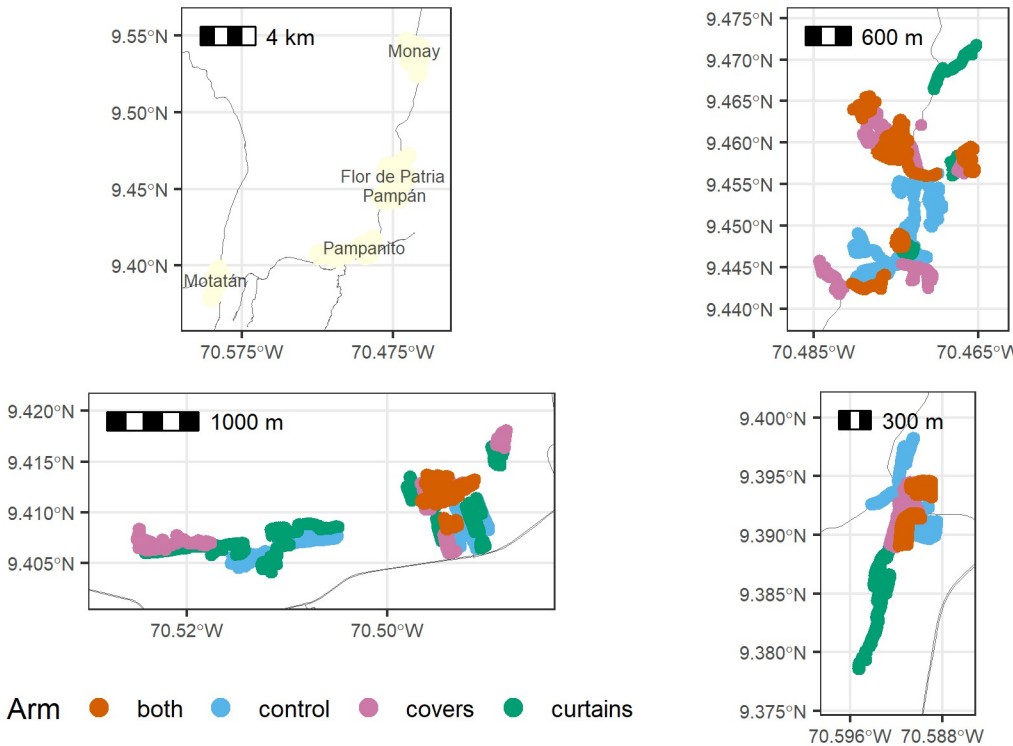

**Fig 1. Maps of the study site.** Top left panel: complete study site. Monay is the external control area, and the other four areas, in which clusters were randomized, are shown in the other three panels. Top right: Flor de Patria and Pampán. Bottom left: Pampanito. Bottom right: Motatán. Solid lines are roads. The colour coding of clusters shows clusters allocated to jar covers only, curtains only, both jar covers and curtains, or control. The base map includes roads from Open Street Map (www.openstreetmap.org) accessed via the R package "osmdata".

intervention), April 2007 (9 months post-intervention), October 2007 (14 months post-intervention), April 2008 (20 months post-intervention) and October 2008 (26 months post-intervention). The trial ended after completion of all follow-up surveys.

## Entomological surveys

After informed consent was obtained from authorities and communities, baseline entomological surveys were conducted in July 2006 in all clusters. Entomological surveys were undertaken in all households to inspect for the presence of *Ae. aegypti* larval habitats. Classical larval surveys [26,27] were used to calculate the Breteau index (BI; number of containers with immature stages per 100 houses), house index (HI; number of houses containing immature stages per 100 houses) and container index (CI; number of containers with immature stages per 100 containers with water). Pupal surveys [28] were used to count the exact number of pupae per positive container and to calculate the pupae per person index (PPI; number of pupae collected/ human population in a cluster). Larval samples were collected from positive containers and taken back to the laboratory at the Universidad de los Andes in Trujillo for species identification. Oviposition traps [29] were placed inside and outside of 10–30% of the houses in each arm, and were left *in situ* for 1 week during each survey period. The primary outcome indicators were the BI and PPI. Secondary indicators were the HI, CI and ovitrap indices. Subsequent entomological surveys were conducted 1-month after the ITMs were distributed and then at 6-monthly intervals until October 2008, for a total of 5 post-distribution entomological

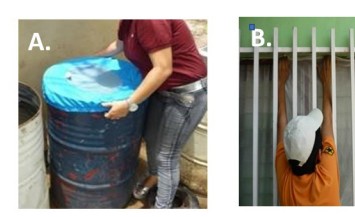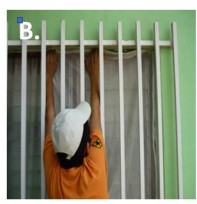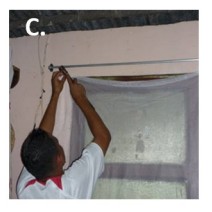

**Fig 2. Photographs show: A) an ITJC being fitted to a water storage drum; B-C) ITCs being hung at windows.**

surveys. BG-Sentinel traps (Biogents AG, Germany) were originally deployed to monitor the adult mosquito populations in a small subset of houses in each study arm. However, due to persistent problems with the electrical supply and householder acceptability, the data collected was incomplete and their use was abandoned after the first follow-up survey.

## ITM interventions

Immediately following the baseline entomological survey, houses in the intervention arms received ITCs, ITJCs or both ITCs + ITJCs. ITCs were PermaNet curtains (Vestergaard-Frandsen, Switzerland), which were factory treated with a long-lasting formulation of deltamethrin (55 mg/m$^2$). This particular PermaNet material had an additional factory treatment with a UV protectant (as many curtains would be exposed regularly to direct sunlight), and the combined UV and insecticidal efficacy was expected to last 18 months or 3–4 washes (personal communication, Vestergaard-Frandsen). The curtains were a standard white colour and measured 1m x 1m. The curtains were hung in all windows regardless of the presence of other window coverings (Fig 2).

ITJCs were provided as ready-to-use products (Vestergaard-Frandsen, Switzerland) and were a pre-packaged, standard size with an elasticated border to close around the water container rim (Fig 2). Households were provided with sufficient ITJCs to cover all large containers used for long-term water storage. These were almost exclusively 150–200 litre drums, and previous research had demonstrated that these containers were the most important to *Ae. aegypti* production in Trujillo [9,30].

Coverage of the ITMs was recorded during every entomological survey.

## Serological surveys

Two serological surveys were undertaken to assess recent dengue infection. The first was conducted nine months after the ITMs were distributed (April 2007) and the second, 32 months after distribution (April 2009). All households with children under the age of 8 were invited to participate. Written informed consent was obtained from the head of household and a blood sample was obtained from one child under 8 years of age in each participating house. The blood sample was collected via finger capillary puncture and was blotted onto an individual piece of filter paper (Whatman). All samples were analysed by dengue IgM capture Enzyme Linked Immunosorbent Assay (ELISA) at Mahidol University's Centre for Vaccine Development in Bangkok, Thailand.

## Insecticide efficacy bioassays

There are currently no specific guidelines for evaluating long-lasting ITMs as curtains against *Ae. aegypti*, so the WHOPES cone bioassay protocol [31] was adapted, substituting *Anopheles*

with *Ae. aegypti* from the insecticide-susceptible Rockefeller laboratory strain. Three cones were placed on separate areas of the curtain, and ten non-blood fed, 2–5 day old *Ae. aegypti* females were introduced into each cone and exposed for three minutes. Mosquitoes were then removed and placed into holding cups, and knockdown was recorded after 1 hour. Mosquitoes were provided cotton soaked in sugar solution and maintained under insectary conditions (23 +/- 2˚C and 70% relative humidity) for 24 hours, when mortality was recorded. WHO criteria were used to categorize bioassay results: 98–100% mortality = susceptible, 90–97% = suspected resistant, <90% = resistant.

Bioassays were performed on curtains collected at baseline (n = 26) and at 6 months (n = 16), 12 months (n = 19) and 24 months (n = 23) after they were delivered in the field. Curtains for the bioassays were collected from houses selected by a computer-generated random list and were replaced with new curtains. Information was collected regarding whether the curtains were exposed to direct sunlight and how often they had been washed. New, unused curtains were used as positive controls.

## Insecticide susceptibility bioassays

*Aedes aegypti* eggs were collected in ovitraps deployed throughout the study area at baseline and at 9, 14, 20 and 26 months after ITM distribution. These eggs were pooled based on study arm and reared to adults under insectary conditions. Adult mosquito insecticide susceptibility bioassays were conducted according to the standard World Health Organization methodology [31], using WHO bioassay tubes and deltamethrin-impregnated papers. Five groups of 15 to 20 unfed females (aged from 1 to 3 days old) were exposed to deltamethrin for 1 hour, and knockdown was recorded at 10, 15, 20, 30, 40, 50 and 60 minutes. After completing the exposure period, the mosquitoes were transferred to recovery chambers and provided with cotton soaked in sugar solution and were maintained under insectary conditions for 24 hours when mortality was recorded. The bioassays were performed in triplicate, with 5 replicates for each population under evaluation. *Aedes aegypti* from the susceptible Rockefeller reference strain were used as a control.

## Data analysis

Data were exported from a custom Microsoft Access database and analysed using R statistical analysis software version 4.1 [32]. For each index and cluster, the total of positive units, over all post-baseline surveys, was calculated to estimate the overall effect, along with the corresponding denominator: e.g., for the Breteau index, the total number of containers with immature stages over the number of houses examined. Then, to compare between arms, a negative binomial regression model was used, with logarithmic link function and the logarithm of the denominator as an offset [33]. This analysis yields ratios of between-arm means. This was preferred over analysis of areas under the curve [14] because the results were highly skewed, yet consistently negative clusters precluded log transformation. A two-sided significance level of 0.05 was used. The same method was used to look for possible systematic differences at baseline between the entomological indices of the external control and the randomized arms. For the randomized arms, application of hypothesis tests would have been logically inconsistent because they would test whether any differences were due to chance, although the randomization ensured that they *were* due to chance. Instead, as a secondary analysis, we follow the recommendation of Altman & Doré [34] and of Senn [35] to include baseline values as a covariate in the regression model. More specifically, we used cluster-level quartiles of the baseline values as a categorical variable in each model.

Coverage of the interventions was defined per house as follows: in the ITC arm, as the proportion of houses with at least 1 ITC observed hanging at the time of the visit; in the ITJC arm, as the proportion appropriately using an ITJC at the time of the visit, with the denominator restricted to those with at least one drum; and in the combined ITC+ITJC arm, as the proportion using at least 1 of the ITMs at the time of the visit. Since coverage was measured after randomization, including it in statistical analysis of the endpoints is likely to be misleading [36]. For example, high indices may result from low coverage of an intervention, or the reverse: poor performance of an intervention may cause trial participants to discontinue use. Some insight may be obtained from looking at temporal patterns of coverage and the indices, but this is complicated by the simultaneous variation in factors such as rainfall. Hence we report only descriptive analyses for coverage.

## Results

### Baseline data

At baseline, 6223 households were recruited into the longitudinal study of entomological indices. Participation fell over the course of the study, and by the final survey, 4504 households (72% of the original cohort) permitted access for entomological surveys to be conducted (Fig 3). No clusters were lost to follow-up.

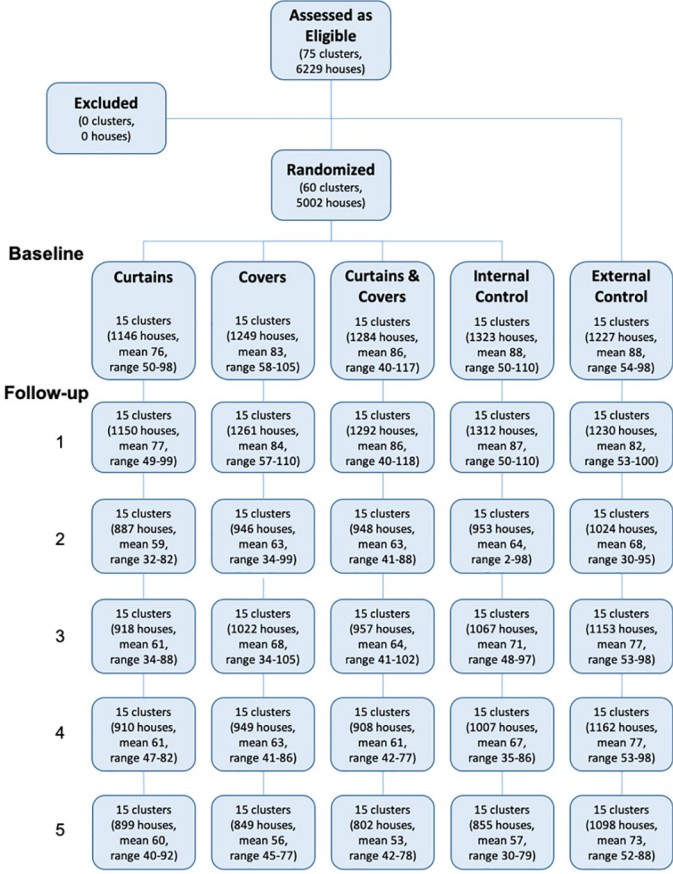

**Fig 3. CONSORT flowchart showing the recruitment and allocation of houses to the study arms and retention over time.**

**Table 1. Mean cluster-level values of the Breteau, Pupae per Person (PPI), House and Container Indices recorded at baseline in the study arms and external control arm, in Trujillo, Venezuela.**

| Arm | Breteau | PPI | House | Container |
|---|---|---|---|---|
| ITCs | 24.93 | 0.19 | 10.71 | 8.62 |
| ITJCs | 19.78 | 0.15 | 10.25 | 6.59 |
| ITC+ITJC | 27.50 | 0.33 | 12.67 | 9.79 |
| Internal Control | 8.87 | 0.14 | 7.24 | 4.32 |
| External Control | 24.77 | 0.32 | 16.97 | 10.51 |

At baseline, a total of 12,564 water-holding containers were inspected in all clusters and were categorised as follows: drums (150–200 litre capacity), small (<10 litre), medium (10–25 litre) and large (>25 litre) containers, tanks, tyres and 'other'. In total, 1023 of these containers (8.1%) were positive for immature *Ae. aegypti*. In terms of vector production, drums were by far the most important larval habitat, confirming earlier surveys in Trujillo [30].

Drums were the most abundant type of water-holding container, accounting for 51.9% (n = 6521) of all containers found, and the most common *Ae. aegypti* larval habitat, comprising 65.7% (n = 672) of the total, followed by small containers (15.4%; n = 158) and tyres (6.1%; n = 62). Of all positive containers, 53.3% (n = 545) were positive for pupae, with drums comprising the greatest proportion of pupae positive containers (62.0%; n = 338), followed by small containers (14.1%; n = 77) and tyres (7.2%; n = 39). Samples from drums yielded 74.3% (10,424/14,028) of the total number of pupae collected in all samples at baseline.

The baseline values of the entomological indices are shown in Table 1 and Fig 3. The values in the external (non-randomized) control arm were higher than in the randomized arms taken together, by factors which ranged from 2.5 (for BI and HI) to 5.3 (for PP), comparing the cluster-level means, and all with p-values less than 0.001.

## Impact of ITMs on the vector

Summarising all survey data post-baseline, the indices in the combination treatment arm (ITCs + ITJCs) were reduced by approximately 50% relative to the internal control (Table 2), differences that were statistically significant for CI and PPI, and borderline for BI. More specifically, the combination treatment reduced CI by 50% (95% confidence interval (95%CI) 13–71%, p = 0.013) and PPI by 63% (95%CI 16–84%, p = 0.016). Relative to the internal control, we observed no index reductions in the ITC arm. Average indices in the external control clusters were at least double the internal control (Table 2), and while these differences are highly statistically significant, the interpretation is limited by the external control not being randomized. Over time, for all the indices, the largest differences between the combination arm and the other treatment groups occurred in the second post-baseline survey (April-May 2007, Fig 4), 9 months post-intervention.

With the onset of the wet season 14 months post-intervention (October 2007), *Ae. aegypti* populations increased substantially in the external control clusters yet remained low in the ITM intervention and internal control arms (Fig 4). However, shortly before the next survey was conducted in April 2008 (20 months post-intervention), a rigorous emergency dengue vector control campaign was mounted by the local health authorities in response to a recent dengue outbreak. That program included space spraying, larviciding and intensive education and container clean-up campaigns and was delivered across a wide area encompassing the trial site. As a result, reductions in all indices were recorded throughout the trial site,

**Table 2. Comparison of entomological indices between trial arms.** The randomized arms (Control, ITC, ITJC and ITC + ITJC) were compared in a single generalized linear model analysis for each endpoint. The non-randomized external control was compared with the randomized control in a separate analysis for each endpoint.

| | Mean | (range) | Ratio versus control (95% CI) p | |
|---|---|---|---|---|
| | | | Primary analysis (unadjusted) | Baseline value of the index as a covariate |
| **Breteau Index (%)** | | | | |
| Control | 11.7 | (1.1–40.1) | - | |
| ITC | 11.1 | (3.6–23.1) | 0.95 (0.52–1.74) 0.86 | 1.02 (0.59–1.76) 0.95 |
| ITJC | 8.6 | (0–23.9) | 0.74 (0.40–1.35) 0.32 | 0.63 (0.38–1.07) 0.094 |
| ITC + ITJC | 6.5 | (0.3–24.5) | 0.55 (0.30–1.01) 0.053 | 0.43 (0.25–0.76) 0.003 |
| External control | 24.2 | (9.3–42.5) | 2.04 (1.24–3.37) 0.005 | |
| **Container Index (%)** | | | | |
| Control | 4.4 | (0.6–14) | - | |
| ITC | 4.8 | (1.1–7.9) | 1.08 (0.62–1.88) 0.78 | 1.10 (0.64–1.87) 0.73 |
| ITJC | 3.1 | (0–8.6) | 0.71 (0.41–1.22) 0.22 | 0.68 (0.41–1.14) 0.16 |
| ITC + ITJC | 2.2 | (0.2–7.3) | 0.5 (0.29–0.87) 0.013 | 0.47 (0.27–0.80) 0.004 |
| External control | 11.5 | (4.3–21.2) | 2.59 (1.64–4.09) <0.0001 | |
| **House Index (%)** | | | | |
| Control | 7.1 | (1.1–22.4) | - | |
| ITC | 7.3 | (2.7–15.3) | 1.03 (0.59–1.78) 0.92 | 0.96 (0.62–1.48) 0.84 |
| ITJC | 4.7 | (0–15.1) | 0.67 (0.39–1.17) 0.16 | 0.55 (0.36–0.86) 0.009 |
| ITC + ITJC | 4.5 | (0.3–18.1) | 0.64 (0.37–1.11) 0.11 | 0.50 (0.31–0.81) 0.004 |
| External control | 15.1 | (6.4–26.6) | 2.10 (1.38–3.2) 0.0006 | |
| **Pupae per person** | | | | |
| Control | 0.11 | (0–0.41) | - | |
| ITC | 0.15 | (0.02–0.42) | 1.38 (0.61–3.12) 0.44 | 1.87 (0.79–4.34) 0.14 |
| ITJC | 0.08 | (0–0.22) | 0.75 (0.33–1.70) 0.49 | 0.73 (0.33–1.66) 0.44 |
| ITC + ITJC | 0.04 | (0–0.13) | 0.37 (0.16–0.84) 0.016 | 0.40 (0.16–0.97) 0.027 |
| External control | 0.51 | (0.16–0.96) | 4.6 (2.04 10.5) 0.0002 | |

including in the external control clusters, and it was not possible to distinguish the impact of the intervention from that of the outbreak response. However, the emergency vector control campaign was short-lived and these drops were not sustained, and by the next and final entomological survey (October 2008; 26 months post-intervention) all four entomological indices

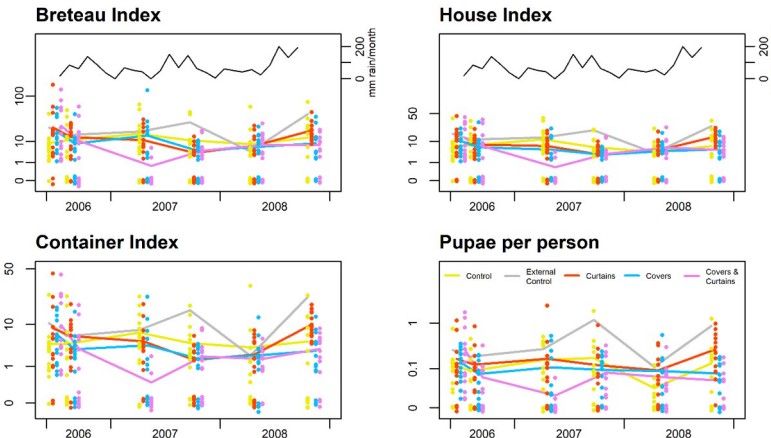

**Fig 4. Entomological indices presented by study arm over time.** Individual dots represent individual clusters. The lines join the arithmetic means over clusters at each time point. The numerical values are included in S1 Table.

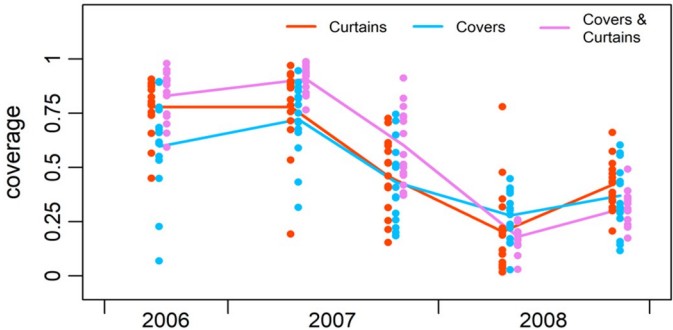

**Fig 5. Coverage of the interventions by study arm over time.** Individual dots represent individual clusters.

were again substantially higher in the external control compared to all three intervention arms (p<0.001).

In summary, ITCs in combination with ITJCs exerted substantial impact on entomological indices. For most of the duration of the trial, indices in the external control arm remained substantially higher than in the ITM study arms (including the internal control), suggesting that the ITMs may have exerted a community effect.

No trends or patterns were detected in analyses of the ovitrap data, for all comparisons within and between study arms.

## Coverage of the interventions

Acceptance of the interventions by households was high. One month after distribution, the average cluster-level coverage in the ITC arm was 78% of households, 62% in the ITJC arm, and 82% of houses using at least 1 ITM in the combination ITC+ITJC arm, with coverage levels rising to 75% in the ITJC and 91% in the combination arm by 8-months post-intervention (Fig 5). However, by 14 months post-intervention (October 2007), coverage levels had fallen to between 44 and 59% across these three arms. The ITJCs experienced widespread physical deterioration, and many people had removed the ITCs to be washed and had not re-hung them. To address this, replacements for damaged or missing ITJCs were offered at the time of the entomological survey in April 2008 (20 months post- intervention). While this resulted in a slight recovery of coverage by the end of the follow-up period (26 months post-intervention), coverage levels never reached their previous high. This drop in coverage might help explain the increase in indices in the combination arm between the second and third post-intervention entomological surveys, although the other indices remained steady.

## Serological results

A total of 8.2% (85/1039) of blood samples were positive for dengue IgM in the first serological survey (April 2007; 8-months post-intervention). Prevalence was consistent across study arms, ranging from 7.5–11.9%, but was notably lower in the external control arm (3.2%). Only 1% (6/783) of blood samples in the second serological survey (April 2009) were positive for dengue IgM, with two positive samples coming from the ITC+ITJC arm, one positive sample from the ITJC arm and three positive samples from the external control.

## Insecticide efficacy bioassays

*Ae*. *aegypti* mortality 24 hours after exposure to new ITCs at baseline varied from 45–100% (mean ± SD: 83% ± 17). Bioassays conducted throughout the course of the study showed

similar results, and after 24 months of field use, the mean 24-hour mortality was 74% (range: 48–97%).

## Insecticide susceptibility bioassays

At baseline, *Ae. aegypti* collected from the three arms under intervention with ITMs as well as the internal control arm were classified as 'suspected resistant' to deltamethrin per WHO guidelines [31], and their status remained consistent throughout the course of the study, with 24-hour post-exposure mortalities ranging from 86–96%. *Ae. aegypti* collected from the external control arm were classified as susceptible for the duration of the study, with 24-hour post-exposure mortality consistently at 99%.

## Discussion

The results showed a reduction in entomological indices in the study arms after the ITMs were deployed, with the indices in the ITC+ITJC arm remaining particularly suppressed over time as compared to the internal control arm. These drops were also observed with respect to the external control arm, suggesting that the ITMs may have exerted a community-wide effect. This trend was most prominent at 14 months post-intervention; while rains peaked and entomological indices surged in the external control site, indices in the study arms, including the internal control, remained remarkably low, even with a coverage of ITMs of about 50%. Entomological data collected 20 months post-intervention were likely impacted by an intensive dengue vector control campaign carried out by the local health authorities, in response to a recent dengue outbreak. However, these activities were not sustained, and entomological indices in the external control arm had dramatically increased again by the next entomological survey. Despite drops in coverage and physical deterioration of the ITMs, entomological indices in the study arms remained significantly lower than in the external control arm at the end of the study. While these findings are similar to those observed in previous ITM studies [9,10], the nature of the present study has allowed for a more comprehensive and longer-term assessment of ITMs for dengue vector control.

Although their efficacy had been previously demonstrated in Mexico by Kroeger *et al.* [9], ITCs alone exerted no notable overall effect on entomological indices in the present study, and indices in the ITC arm recovered to baseline levels by the end of the study. However, at the end of the study, the values of all 4 entomological indices in the ITC arm remained significantly lower than in the external control. This difference might suggest that the ITC arm may have benefited from the protective efficacy of the interventions in other study arms, with the difference attributed partly to the community effect and partly to the retained efficacy of the ITCs still in place. Bioassays indicated that the ITCs were functioning at acceptable levels of insecticidal efficacy, but coverage had dropped dramatically by 14-months. A complementary study of ITM effectiveness in a neighbouring part of Trujillo State also found a coverage-dependent impact of a combined ITC+ITJC intervention, which led to significant reductions in BI and PPI, although coverage declined dramatically over the 18-month study period [11]. Further research by Vanlerberghe *et al.* [37] also concluded that reductions in BI and PPI in an area that received ITCs in Thailand were heavily coverage-dependent. Despite the best efforts of the field team, coverage of the interventions never returned to the levels seen at the outset of the study, suggesting that the long-term use of ITMs by householders may be one of the greatest challenges to the success of their application.

Although coverage with ITJCs was also low over the long-term, the differential impact of the intervention was likely greater than that of the ITCs alone, but overall was not significant in comparison to the internal control. In addition to providing a mechanical barrier to

oviposition, the insecticide on the covers should have killed any females immediately after contact, thus denying them the chance to oviposit elsewhere. Prior to intervention, drums had produced 74% of the standing crop of pupae, so even if only 30–40% of the drums were correctly using the ITJCs, significant reductions in mosquito abundance could reasonably be expected. Hence, indices remained suppressed in the ITJC arm for the duration of the study, and when the ITJCs were combined with ITCs, the impact on entomological indices was even more pronounced.

The results reported by Kroeger *et al.* [9] for the combined ITC+ITJC intervention previously implemented in Trujillo town are similar to those reported here, showing an initial and sustained suppression of indices for the duration of the study. In the present study, the impact on entomological indices was most evident in the combined ITC+ITJC arm. This outcome is not entirely unexpected, as the households in the ITC+ITJC arm benefited from a reduction in the availability of preferred larval habitats and an increase in surface area treated with residual insecticide provided by the ITCs in the windows. Whereas the follow-up period in the previous study in Trujillo had been 9-months, the present study evaluated the efficacy of the ITMs over a 26-month period, demonstrating that ITMs are able to suppress dengue vector populations over the long term, even when coverage is suboptimal, particularly when ITCs and ITJCs are deployed in combination.

The development of insecticide resistance is always a concern in any insecticide-based vector control intervention. It was encouraging to observe that the ITMs did not affect the susceptibility of the local *Ae. aegypti* population to deltamethrin, despite baseline evidence that the population was considered at risk for the development of resistance.

In the results presented, there are two important shortcomings despite the considerable effort made carrying out the sampling procedures to collect the data. First, there is no accurate assessment of the impact of the ITM interventions on the adult mosquito populations. The monitoring of adult *Ae. aegypti* had originally been planned using BG-Sentinel traps, but the traps did not function efficiently during the early stages of the trial. This was primarily due to persistent night-time power outages. In the few instances where the traps should have functioned optimally, the catches were low and it was suspected that, in some cases at least, the householders may have disconnected the traps due to concerns about their electrical consumption. This is unfortunate as it would have been instructive to compare the performance of these loosely but rapidly fitted ITMs with that of similar ITMs when tightly-fitted as window screens [18].

The second shortcoming is the inconclusive nature of the serological data collected to measure the impact of ITMs on dengue transmission. The dengue IgM ELISA was chosen to assess recent dengue infection in children under 8 years of age, under the assumptions that they would be less likely to have previously had dengue and would most likely be exposed to dengue in the home environment. However, dengue transmission intensity tends to vary widely between years, and in the post-intervention survey, overall sero-prevalence was much lower than in the first survey. While dengue incidence appeared to be stable throughout the first year of the study, the numbers of reported cases in Trujillo State increased by up to 40% soon after introduction of the ITMs in April 2007, with levels sustained through the first ten epidemiological weeks of 2008 (Dirección Regional de Epidemiología y Estadística Vital, Trujillo-Venezuela, 2008). Hence, given the low number of cases and insufficient power of the study, no comparison could be made between study arms.

In conclusion, this study adds to the body of evidence on ITMs for control of *Ae. aegypti* by demonstrating their potential to exert a long-term impact on vector populations, a considerable benefit for routine dengue vector control operations. In the event of an outbreak or in anticipation of a seasonal rise in transmission, ITMs could be deployed rapidly as done in the

present study. The global challenge of pyrethroid resistance in *Ae. aegypti* could be met by potentially using netting treated with non-pyrethroid insecticides. However, efficacy of ITMs is highly dependent not only on levels of coverage and on the local characteristics of dengue epidemiology and accompanying control measures, but also on the suitability of the houses in the intervention area. While the impact of ITMs on immature mosquito populations is compelling, further research is required to assess their implementation cost, their impact on adult vector populations and, most crucially, on dengue transmission.

## Supporting information

**S1 Table. Cluster-level means of each index in each arm at each survey. a**. Pupae/person Index summary. **b**. House Index Summary. **c**. Container index Summary. **d**. Breteau Index Summary.
(DOCX)

## Acknowledgments

The authors would like to thank Leslie Alvarez, Milagros Oviedo, and Nelson Carrillo for their assistance and the members of the study communities for their participation. We are indebted to Prof. Sutee Yoksan of the Centre for Vaccine Development, Institute of Molecular Biosciences, Mahidol University, Bangkok, Thailand, for performing the serological analyses.

## Author Contributions

**Conceptualization:** Audrey Lenhart, Neal Alexander, Patrick van der Stuyft, Philip J. McCall.

**Data curation:** Carmen Elena Castillo, Neal Alexander.

**Formal analysis:** Audrey Lenhart, Carmen Elena Castillo, Neal Alexander.

**Funding acquisition:** Audrey Lenhart, Elci Villegas, Neal Alexander, Veerle Vanlerberghe, Patrick van der Stuyft, Philip J. McCall.

**Investigation:** Audrey Lenhart, Carmen Elena Castillo, Veerle Vanlerberghe.

**Methodology:** Audrey Lenhart, Neal Alexander, Patrick van der Stuyft, Philip J. McCall.

**Project administration:** Audrey Lenhart, Elci Villegas, Patrick van der Stuyft, Philip J. McCall.

**Resources:** Audrey Lenhart, Carmen Elena Castillo, Veerle Vanlerberghe.

**Supervision:** Audrey Lenhart, Carmen Elena Castillo, Elci Villegas, Patrick van der Stuyft, Philip J. McCall.

**Validation:** Audrey Lenhart.

**Writing – original draft:** Audrey Lenhart, Carmen Elena Castillo, Neal Alexander, Philip J. McCall.

**Writing – review & editing:** Audrey Lenhart, Carmen Elena Castillo, Elci Villegas, Neal Alexander, Veerle Vanlerberghe, Patrick van der Stuyft, Philip J. McCall.

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
