## [Decision Letter · Decision Letter 0]

5 Aug 2021

Dear Prof McCall,

Thank you very much for submitting your manuscript "Evaluation of insecticide treated window curtains and water container covers for dengue vector control in a large-scale cluster-randomized trial in Venezuela" for consideration at PLOS Neglected Tropical Diseases. As with all papers reviewed by the journal, your manuscript was reviewed by members of the editorial board and by several independent reviewers. The reviewers appreciated the attention to an important topic. Based on the reviews, we are likely to accept this manuscript for publication, providing that you modify the manuscript according to the review recommendations. 

Sincerely,

Guilherme S. Ribeiro, M.D., M.Sc., Ph.D

Associate Editor

Nigel Beebe

Deputy Editor

Reviewer's Responses to Questions

**Key Review Criteria Required for Acceptance?**

**Methods**

-Are the objectives of the study clearly articulated with a clear testable hypothesis stated?

-Is the study design appropriate to address the stated objectives?

-Is the population clearly described and appropriate for the hypothesis being tested?

-Is the sample size sufficient to ensure adequate power to address the hypothesis being tested?

-Were correct statistical analysis used to support conclusions?

-Are there concerns about ethical or regulatory requirements being met?

Reviewer #1: This review is by Carlos Chaccour, ISGLobal, Barcelona Institute for Global Health. It is my personal policy to conduct open reviews.

This manuscript describes and reports a cluster randomized trial that assesses insecticide treated curtains and water jar covers to reduce Dengue transmission using entomological and serological outcome measures.

The methods are appropriate and the sample size calculations are fitting. The manuscript however refers to repeatedly to a spillover effect and uses and external control to partly assess that. This is all very well but it is difficult to assess without a clear description of the distance between clusters and whether or not buffer zones were included and received the intervention. An explicit description of the cluster design i.e. fried egg vs full gradient vs partial gradient would be very helpful in understanding the spillover. I would suggest including an schematic or a map with this information. 

Data analysis: what is meant by "by definition, the randomized arms did not differ systematically at baseline" on page 7?

Reviewer #2: The objectives of the study are clearly articulated with study design. 

I propose the following points be addressed. 

Study area

- The authors could provide more general information about the breeding sites and productivity of Ae. aegypti, characteristics of water containers, house structure suitable for ITM deployment, as well as other socioeconomic characteristics and incidence of dengue to better describe and characterize the context of the study area. 

Entomological surveys

- Interventions were deployed in households and entomological surveys were conducted on households. Cluster Public spaces surrounding households also important places to be assessed, mosquitoes tend to be displaced from households to harbor other places, when principal breeding sites are intervened. Entomological surveys in public spaces were conducted? Do authors consider this information adds valuable information?

- Authors state: “BG Sentinel traps (Biogents AG, Germany) were originally deployed to monitor the adult mosquito populations in a small subset of houses in each study arm. However, due to persistent problems with the electrical supply and householder acceptability, the data collected was incomplete and their use was abandoned after the first follow-up survey”. Even though this method was abandoned I suggest mention the number of households that represent the small subset. Why when facing this barrier an alternative method for adult collection was not used? 

Serosurveys

- The authors define that “Two serological surveys were undertaken to assess recent dengue infection. The first was conducted nine months after the ITMs were distributed (April 2007) and the second, 32 months after distribution (April 2009). All households with children under the age of 8 were invited to participate. Written informed consent was obtained from the head of household and a blood sample was obtained from one child under 8 years of age in each participating house. The blood sample was collected via finger capillary puncture and was blotted onto an individual piece of filter paper (Whatman)” Please clarify : 

• The first serological survey was conducted nine or eight months after ITMs distribution? Methods section mentions nine months while, results section 8 months. 

• Why a baseline serological survey was not carried out?

• What was the sample size calculated for this outcome?

- “All samples were analyzed by dengue IgM capture ELISA at Mahidol University’s Centre for Vaccine Development in Bangkok, Thailand”.: 

• Suggest to write “enzyme linked immunosorbent assay” before ELISA

• How results were classified: positive, negative or borderline?

• What is the value of cutoff?

• Which kit was used? Reagents? Reference?

Insecticide efficacy bioassays

- Was Insecticide efficacy bioassays conducted with ITJC ? ITJC are exposed to other factors that may influence insecticide efficacy.

Insecticide susceptibility bioassays

- Please clarify in method section how the results were categorized ? Results section indicates “suspected resistant” 

Data analysis

- Citation of R statistical package is missing. I suggest citation as follows: R Core Team (2020). R: A language and environment for statistical computing. R Foundation for Statistical Computing, Vienna, Austria. URL https://www.R-project.org/

- Randomization occurred among intervention and control clusters and arms did not differ systematically at baseline, but the study design is longitudinal and other intermediate factors may influence the effect. 

• Did the negative binomial regression model used considered possible intermediate variables? For example: climate, household variables or container variables that change over time, ITM conditions (exposition to sunlight, number of washes during the study period, damages, % of holes, coverage? 

• Please clarify if the effect of the intervention was adjusted for other intermediate variables that may be influencing the effect and it is not only due to the intervention. 

Coverage

- “Coverage of the interventions was defined per house as follows: in the ITC arm, as the proportion of houses with at least 1 ITC observed hanging at the time of the visit; in the ITJC arm, as the proportion appropriately using an ITJC at the time of the visit, with the denominator restricted to those with at least one drum; and in the combined ITC+ITJC arm, as the proportion using at least” 

• Time of the visit includes baseline?

• What is the exact number of ITC and ITJC deployed per cluster? Per household?

• Was “poor performance of the intervention” measured? 

• Other variables of use, uptake and satisfaction of intervention measured? This will add information regarding recommendations about coverage. Some information is given in results section. O suggest to clarify the descriptive analysis carried out stating the variables presented in results section. 

Ethical approval

- In some countries IRBs and local regulations require that young subjects how can read signed an assent form in addition to parental consent form. Is this a requirement in Venezuela ? If so please clarify.

Reviewer #3: Objectives and Methods are correct and appropriate

**Results**

-Does the analysis presented match the analysis plan?

-Are the results clearly and completely presented?

-Are the figures (Tables, Images) of sufficient quality for clarity?

Reviewer #1: I suggest including statistical testing of the entomological measurements at baseline presented in table 1.

Reviewer #2: Analysis presented is in accordance with the analysis proposed in methods. 

I suggest authors clarfy or add 

Serological results

- Interestingly if some Dengue incidence data is described during study periods

- The serological surveys were carried 9 months or 8 months after intervention implementation?

- Range of age, mean age of among study arms and surveys are not reported.

Coverage 

- In results section: “The ITJCs experienced widespread physical deterioration, and many people had removed the ITCs to be washed and had not re-hung them. To address this, replacements for damaged or missing ITJCs were offered at the time of the entomological survey in April 2008 (20 months post- intervention)”

- How many IT materials were replaced? By damage or missing per type ITC ITJC

Reviewer #3: Analyses and results are correct and appropriate

**Conclusions**

-Are the conclusions supported by the data presented?

-Are the limitations of analysis clearly described?

-Do the authors discuss how these data can be helpful to advance our understanding of the topic under study?

-Is public health relevance addressed?

Reviewer #1: Conclusions are generally supported by the results. Given the large difference between internal and external controls at baseline, I suggest de-emphasizing the community effect or better supporting this interpretation. Particularly in the abstract.

Reviewer #2: Conclusiones and limitations are clearly presented. However authors should clarify the following: 

- 2nd paragraph last sentence reads as: “Despite the best efforts of the field team, coverage of the interventions never returned to the levels seen at the outset of the study, suggesting that the long-term use of ITMs by householders may be one of the greatest challenges to the success of their application”

• Authors may add some further recommendations to assess this point. For example: should factors associated to use be assessed in depth. Have other studies assessed this?

- “The second shortcoming is the inconclusive nature of the serological data collected to measure the impact of ITMs on dengue transmission. The dengue IgM ELISA was chosen to assess recent dengue infection in children under 8 years of age, under the assumptions that they would be less likely to have previously had dengue and would most likely be exposed to dengue in the home environment. However, dengue transmission intensity tends to vary widely between years, and in the post-intervention survey, overall sero-prevalence was much lower than in the first survey. Hence, given the low number of cases and insufficient power of the study, no comparison could be made between study arms.

• How did dengue transmission vary during study periods (epidemic or endemic periods)?

Reviewer #3: Authors discuss how data is helpful and the public health relevance of the study. Limitations are also described.

**Editorial and Data Presentation Modifications?**

Reviewer #1: (No Response)

Reviewer #2: (No Response)

Reviewer #3: Only a minor modification is suggested. 

In the Abstract/Conclusions the authors state: “ITMs can SUPPRESS dengue vector populations in the long term, EVEN WHEN COVERAGE IS SUBOPTIMAL, particularly when ITCs and ITJCs are deployed in combination.

I would recommend to the authors to eliminate SUPPRESS and to use another verb/word. Their intervention did not prove to suppress dengue vector populations in the long term.

Also, to change SUBOPTIMAL, unless it is clearly stated in the text what would be an “OPTIMAL” coverage.

**Summary and General Comments**

Reviewer #1: (No Response)

Reviewer #2: Once reviewed the paper Evaluation of insecticide treated window curtains (ITM) and water container covers for dengue vector control in a large-scale cluster-randomized trial in Venezuela, I consider the work relevant and manuscript is worthy of publication. This study was well conducted and it is an important contribution to other evaluations of the effectiveness of ITM as an intervention to control mosquitoes. However, I have minimal suggestions and address several remarks to be clarified.

Abstract: This section is designed to highlight key points from major sections of the paper and to explain what the paper includes. Effective abstracts provide sufficient details to expedite classifying the paper as relevant (or not) to readers' clinical work or research interests, therefore:

1. Additional information is needed to clarify the type of statistical model used.

2. The full text describes as primary outcomes BI and PPI and as secondary outcomes ovitraps however the abstract reports that primary outcomes are: BI, PPI, CI, HI and ovitrap index).

3. I suggest to include the time frame when the study was conducted.

4. Information about both insecticide efficacy and susceptibility is missing. 

Full text

Introduction

1. Second paragraph references are included as [4, 5, 6, 7] better (4-7). Please review writing guidelines,

2. Fourth paragraph [9, 10, 11, 12, 13] better 9-13

Other suggestions were already addressed in the later sections.

Reviewer #3: The authors report a very nice and large study following earlier trials of the efficacy of insecticide treated curtains (ITCs) and water jar covers (ITJCs) for the control of Aedes aegypti. Objectives and Methods are correct and appropriate. Authors discuss how data is helpful and the public health relevance of the study. Limitations are also described.

Only a minor modification is suggested. In the Abstract/Conclusions the authors state: “ITMs can SUPPRESS dengue vector populations in the long term, EVEN WHEN COVERAGE IS SUBOPTIMAL, particularly when ITCs and ITJCs are deployed in combination. I would recommend to the authors to eliminate SUPPRESS and to use another verb/word. Their intervention did not prove to suppress dengue vector populations in the long term. Also, to change SUBOPTIMAL, unless it is clearly stated in the text what would be an “OPTIMAL” coverage.

PLOS authors have the option to publish the peer review history of their article (what does this mean?). If published, this will include your full peer review and any attached files.

Reviewer #1: Yes: Carlos Chaccour

Reviewer #2: No

Reviewer #3: No

Figure Files:

Data Requirements:

Reproducibility:

References

---

## [Decision Letter · Decision Letter 1]

30 Dec 2021

Dear Prof McCall,

We are pleased to inform you that your manuscript 'Evaluation of insecticide treated window curtains and water container covers for dengue vector control in a large-scale cluster-randomized trial in Venezuela' has been provisionally accepted for publication in PLOS Neglected Tropical Diseases.

Best regards,

Guilherme S. Ribeiro, M.D., M.Sc., Ph.D

Associate Editor

Nigel Beebe

Deputy Editor

Reviewer's Responses to Questions

**Key Review Criteria Required for Acceptance?**

**Methods**

-Are the objectives of the study clearly articulated with a clear testable hypothesis stated?

-Is the study design appropriate to address the stated objectives?

-Is the population clearly described and appropriate for the hypothesis being tested?

-Is the sample size sufficient to ensure adequate power to address the hypothesis being tested?

-Were correct statistical analysis used to support conclusions?

-Are there concerns about ethical or regulatory requirements being met?

Reviewer #2: YES

Reviewer #3: (No Response)

**Results**

-Does the analysis presented match the analysis plan?

-Are the results clearly and completely presented?

-Are the figures (Tables, Images) of sufficient quality for clarity?

Reviewer #2: YES

Reviewer #3: (No Response)

**Conclusions**

-Are the conclusions supported by the data presented?

-Are the limitations of analysis clearly described?

-Do the authors discuss how these data can be helpful to advance our understanding of the topic under study?

-Is public health relevance addressed?

Reviewer #2: YES

Reviewer #3: (No Response)

**Editorial and Data Presentation Modifications?**

Reviewer #2: ACCEPT THE MANUSCRIPT AS EDITED

Reviewer #3: (No Response)

**Summary and General Comments**

Reviewer #2: ALL RECOMENDATIONS AND COMENTARIES ANNOTED WHERE CONSIDERED BY AUTHORS

Reviewer #3: The authors have addressed satisfactorily all the points from the previous review.

PLOS authors have the option to publish the peer review history of their article (what does this mean?). If published, this will include your full peer review and any attached files.

Reviewer #2: **Yes: **JULIANA QUINTERO

Reviewer #3: No

---

## [Editor Report · Acceptance letter]

2 Mar 2022

Dear Prof McCall,

We are delighted to inform you that your manuscript, "Evaluation of insecticide treated window curtains and water container covers for dengue vector control in a large-scale cluster-randomized trial in Venezuela," has been formally accepted for publication in PLOS Neglected Tropical Diseases.

Best regards,

Shaden Kamhawi

co-Editor-in-Chief

Paul Brindley

co-Editor-in-Chief
